# Potential of LLM-Generated Lifestyle Adjustment Recommendations Based on Multimodal Data

## Abstract

The prevalence of burnout, depression, and stress-related disorders has increased markedly in contemporary societies, particularly in the context of flexible and remote working arrangements. These structural shifts impose novel demands on individuals to self-regulate health, well-being, and productivity—responsibilities that were previously supported by organizational structures in conventional workplaces. Traditional self-management strategies struggle to address the complexity of interacting behavioral, psychological, and physiological determinants. This paper explores the feasibility of employing large language models (LLMs) to generate lifestyle adjustment recommendations based on multimodal data that integrate subjective self-reports with objective sensor-derived measures. To this end, we simulate realistic multimodal time-series data for a prototypical remote worker, design a natural language prompt to elicit recommendations from an LLM, and employ an independent LLM to evaluate the generated outputs in terms of safety, relevance, and feasibility. The results suggest that LLMs are capable of detecting meaningful behavioral patterns and translating them into actionable guidance. This approach has the potential to support individuals in developing adaptive routines for health and productivity management. Future research should emphasize real-world validation, integration with digital health platforms, and the establishment of ethical safeguards.

## 1 Motivation

The organization of work has undergone profound transformations in recent decades. Increasingly flexible arrangements, hybrid work models, and the widespread adoption of remote working have blurred the boundaries between professional and private life. While these developments afford individuals greater autonomy, they simultaneously increase the demands on self-organization. Many workers report difficulties in maintaining structure, with consequences for sleep quality, physical activity, and mental health. Epidemiological data confirm substantial increases in the prevalence of burnout and depression over the past decade, with implications for individual well-being, workforce productivity, and public health systems (World Health Organization, 2022).

In conventional office settings, organizational routines such as fixed schedules, communal breaks, and social support structures provided external scaffolding for daily rhythms. The dissolution of such anchors in remote and hybrid contexts shifts responsibility for health and productivity management to individuals. However, effective self-management requires balancing multiple interacting factors, including sleep, stress, nutrition, physical activity, and social connectedness. Human introspection is often insufficient to capture these dynamic interrelations, leaving individuals unable to identify subtle but consequential patterns.

Advances in digital health technologies offer promising avenues for addressing this challenge. Wearables and smartphones now enable continuous monitoring of physiology and behavior, while

digital platforms facilitate real-time self-reporting. Nevertheless, a persistent gap remains between the availability of heterogeneous, temporally dense data streams and their translation into actionable insights. LLMs, with their capacity to process natural language and structured data, represent a potential solution. By integrating multimodal data, LLMs could generate personalized recommendations expressed in accessible, everyday language. This paper explores this proposition through a simulated case study.

## 2 Related Work: Diary Studies and Personal Informatics

Psychological and health research has a long tradition of employing diary methods to capture experiences in daily life. Approaches such as the Experience Sampling Method (ESM) and Ecological Momentary Assessment (EMA) allow participants to report moods, behaviors, and contextual factors in real time. These methodologies reduce recall bias and provide ecologically valid insights into fluctuating states [1]. Applications span clinical psychology, organizational behavior, and health research. For instance, Verhagen et al. [2] demonstrated the utility of EMA in psychiatric populations, while Fritz et al. [3] synthesized methodological best practices for intensive longitudinal designs.

Complementing subjective measures, digital phenotyping and personal sensing approaches leverage sensors embedded in smartphones and wearables to capture behavioral and physiological data unobtrusively. Mohr et al. [4] described personal sensing as a paradigm shift in mental health, enabling objective assessment of activity, sleep, and social interactions. Pizzoli et al. [5] reviewed advances in digital phenotyping, highlighting the utility of accelerometry, GPS, and app usage data as proxies for lifestyle behaviors. Torrado et al. [6], in the ActiveAgeing study, demonstrated the value of combining wearable sensors with qualitative methods to monitor older adults.

Although diary methods and personal informatics yield rich insights into daily life, interpretation typically remains the responsibility of researchers or individuals themselves. Ecological momentary interventions [7] have sought to deliver just-in-time feedback, yet the integration of multimodal data into automated, personalized guidance remains limited. This gap underscores the opportunity for LLMs to serve as intermediaries between raw data and actionable lifestyle advice.

## 3 Analysis of the Potential of LLM-Based Lifestyle Adjustment Recommendations

### 3.1 Multimodal Dataset

To examine feasibility, we simulated a seven-day multimodal dataset for a fictional persona, Alice, a 35-year-old remote software developer. Each day includes self-reported mood, perceived stress, and productivity (scales 1–10), alongside wearable-derived measures: daily step count, mean heart rate variability (HRV), sleep duration, sedentary hours, and self-logged social contacts (interactions >10 minutes with colleagues, friends, or family).

The data indicate clear associations (see Tables 1 and 2): days with insufficient sleep (<6.5 h) correspond to lower mood, higher stress, reduced productivity, diminished HRV, fewer steps, and limited social interaction (Days 2, 3, and 6). Conversely, days with sufficient sleep, higher physical activity, and at least three social contacts (Days 1, 4, and 7) align with improved mood and productivity. These observations highlight the interdependence of sleep, activity, social engagement, and well-being.

### 3.2 Prompt Design

We developed a structured natural language prompt instructing an LLM to analyze Alice's dataset and generate recommendations. The prompt emphasized safety, contextual relevance, and feasibility, requiring outputs to consist of complete sentences with concise justifications.

The following prompt (template) was used:

```
You are given a seven-day dataset from a 35-year-old remote software
developer named Alice.  The dataset includes daily values for mood,
stress, productivity, steps, heart rate variability (HRV), sleep
duration, sedentary hours, and social contacts.
```

Table 1: Summary of subjective measures averaged across stress-level groups

| Group | Mood | Stress | Productivity |
|---|---|---|---|
| High stress (Days 2, 3, 6) | 4.7 | 7.3 | 4.7 |
| Moderate stress (Day 5) | 6.0 | 6.0 | 6.0 |
| Low stress (Days 1, 4, 7) | 6.7 | 4.3 | 7.7 |

Table 2: Summary of objective measures averaged across stress-level groups

| Group | Steps | HRV (ms) | Sleep (h) | Sed. (h) | Contacts |
|---|---|---|---|---|---|
| High stress (Days 2, 3, 6) | 4,167 | 35.0 | 6.0 | 10.3 | 0.7 |
| Moderate stress (Day 5) | 6,100 | 39.0 | 6.9 | 9.0 | 2.0 |
| Low stress (Days 1, 4, 7) | 8,400 | 44.3 | 7.7 | 7.3 | 3.3 |

```
Here is the data:  {DATA}
Your task is to:
 1. Analyze the dataset as a whole, looking for patterns across the
    seven days.
 2. Generate lifestyle adjustment recommendations that are safe,
    contextually relevant, and feasible for someone in Alice's
    situation.
 3. Present your recommendations in complete sentences.
 4. Provide a brief justification for each recommendation,
    explaining which data patterns support it.
 5. Focus only on constructive, non-clinical advice (e.g., daily
    routines, work breaks, activity balance, social habits, sleep
    hygiene).
Avoid medical diagnoses or treatments.  Write clearly and concisely.
```

## 3.3 Example Recommendations

The LLM examined the following data patterns:

1. **Mood, Stress, Productivity:** Mood and productivity are higher, and stress lower, on days with more sleep, higher steps, and more social contacts (Days 4 and 7). Conversely, mood and productivity drop when sleep is short, steps are low, and stress rises (Days 2–3, 6).

2. **Activity & Sedentary Time:** Days with fewer than 5,000 steps and long sedentary hours (10–11 h) correspond to lower mood and productivity.

3. **Sleep:** Better days (4 and 7) align with 7.8–7.9 hours of sleep. Short sleep (5.9–6.2 h) is linked with worse stress and productivity.

4. **HRV:** HRV is higher (45–46 ms) on good days, and lower (34–36 ms) on stressful, less active days.

5. **Social Contacts:** More interactions (3–4 contacts) coincide with better mood and productivity.

It produced the following recommendations:

1. **Prioritize consistent sleep of 7–8 hours per night.** Justification: On days with longer sleep (Days 4 and 7), Alice's mood, stress, and productivity improved. Maintaining this range could help stabilize energy and performance.

2. **Incorporate short movement breaks during work hours to reduce sedentary time.** Justification: Higher sedentary hours (10–11 h) were linked with lower steps, mood, and productivity (Days 2–3, 6). Standing or stretching breaks every hour may help counteract this.

Table 3: Evaluation of lifestyle recommendations on safety, relevance, and feasibility

| Recommendation | Safety | Relevance | Feasibility |
|---|---|---|---|
| Prioritize consistent sleep of 7–8 hours per night | Safe; aligns with sleep guidelines | Data: longer sleep leads to better mood, stress, productivity | Realistic with good sleep hygiene |
| Incorporate short movement breaks during work hours | Safe; light activity, minimal risk | Data: high sedentary leads to lower mood/productivity | Highly feasible (standing, stretching, walking) |
| Aim for at least 7,000–8,000 steps most days | Safe; aligns with activity guidelines | Data: higher steps lead to better mood, HRV, productivity | Achievable, requires planning on busy days |
| Schedule intentional social interactions | Safe; psychologically beneficial | Data: more contacts lead to higher mood, productivity | Feasible, depends on schedule/social network |
| Plan focused work blocks with breaks | Safe; aligns with healthy work practices | Data: lower stress with higher productivity | Feasible with structured planning |

3. **Aim for at least 7,000–8,000 steps most days.** Justification: Days with more steps (Day 4: 8,600; Day 7: 9,100) coincided with higher mood, productivity, and HRV. Increasing daily activity appears beneficial.

4. **Schedule intentional social interactions throughout the week.** Justification: Social contact was highest (3–4) on Alice's best days for mood and productivity. Regular connections may provide emotional balance and reduce stress.

5. **Plan focused work blocks with breaks to manage stress.** Justification: Stress was lowest when productivity was high (Days 4 and 7). Structured routines that mix deep work and recovery breaks could replicate these conditions more often.

## 3.4 Evaluation by a Secondary LLM

A secondary LLM evaluated the recommendations for safety, relevance, and feasibility (Table 3). All were rated safe for a healthy adult, directly grounded in the observed patterns, and realistic to implement. The evaluator further noted consistency with established psychological evidence, particularly regarding the benefits of sleep, activity, and social support.

For evaluation, following prompt template was used:

```
You are given a set of lifestyle recommendations generated for a
healthy adult based on a seven-day dataset.
Your task is to evaluate each recommendation on three dimensions:
  1. Safety - Is the recommendation safe for a healthy adult?
  2. Relevance - Is it directly grounded in the observed data
     patterns?
  3. Feasibility - Is it realistic to implement in daily life?
Provide a brief judgment for each dimension, using complete
sentences.  If applicable, note whether the recommendation is
consistent with established psychological or behavioral evidence
(e.g., benefits of sleep, activity, or social support).
Here are the recommendations:  {RECOMMENDATIONS}
```

## 3.5 Discussion

We illustrated that LLMs can interpret multimodal datasets encompassing behavioral, physiological, and social variables to generate contextually appropriate recommendations. Importantly, the design of prompts proved critical in constraining outputs toward safe, specific, and actionable suggestions. The inclusion of social interaction metrics expanded interpretive scope beyond physical and physiological

dimensions, underscoring the multidimensional nature of well-being. To extend beyond the proof-of-concept stage, future research should deploy the proposed workflow on real-world multimodal datasets collected through combinations of wearables and ecological momentary assessment (EMA) applications. Even small-scale pilot studies would provide valuable ecological validity, enabling comparison between LLM-derived recommendations and patterns observed in naturalistic settings. Such validation would directly address the limitations of simulation and strengthen the robustness of the approach.

# 4 Limitations

Several limitations of this exploratory work should be acknowledged. The analysis relied on a simulated dataset rather than empirical observations, which allowed for systematic illustration of the proposed workflow but restricted the ecological validity of the findings. Application to authentic multimodal data streams, such as those combining wearable sensing with ecological momentary assessment, will be necessary to establish robustness under naturalistic conditions. The evaluation of recommendations was conducted by a secondary LLM from the same model family as the generator. While this setup allowed scalable and efficient assessment, it risks bias through shared training data and architectural characteristics, meaning that the evaluator may replicate or reinforce the generator's assumptions rather than provide an independent appraisal. While this approach enabled rapid and scalable appraisal of safety, feasibility, and relevance, it cannot substitute for expert or clinical judgment, and recommendations deemed plausible by an LLM may nevertheless prove infeasible or ethically problematic in practice. The scope of the dataset was deliberately narrow, encompassing only a small set of daily variables over a seven-day period, whereas human health and behavior are shaped by a far broader constellation of contextual and longitudinal influences including nutrition, workload, socioeconomic status, and environment. Finally, the present design does not address questions of privacy, personalization boundaries, or user agency. The integration of LLM-generated recommendations into everyday life raises substantive ethical and governance challenges related to accountability, transparency, and informed consent, all of which must be addressed before real-world deployment can be responsibly pursued. Taken together, these constraints delineate clear priorities for future work, including empirical validation, expansion of multimodal scope, and the development of safeguards that enable the responsible integration of LLM-based recommendation systems into digital health practice.

# 5 Conclusion and Outlook

This study explored the potential of LLMs to generate lifestyle adjustment recommendations based on multimodal data integrating subjective and sensor-derived measures. Using a simulated dataset, we demonstrated that LLMs can detect meaningful behavioral patterns and transform them into actionable, safe, and feasible advice. The use of a secondary LLM as an evaluator further suggests pathways for embedding quality control mechanisms into recommendation pipelines.

More broadly, the dual-LLM pipeline—consisting of a generator model that produces candidate recommendations and an evaluator model that rates their safety, feasibility, and contextual fit—could be generalized as a reusable research workflow across domains. Beyond digital health, analogous "generate-then-evaluate" pipelines may support tasks in education, computational creativity, or human–computer interaction research, where iterative AI-based self-checking reduces the risk of unsafe or irrelevant outputs. Adapting the workflow to these domains would require tailoring evaluation criteria to context: for instance, pedagogical appropriateness and inclusivity in education, or novelty and coherence in creativity research. Safeguards would need to be domain-specific but consistently oriented toward ensuring that outputs remain relevant, non-harmful, and ethically appropriate.

Future research should validate these findings in real-world contexts, with naturalistic multimodal datasets and expert benchmarks. Even small-scale pilot studies—for example, involving 15–20 participants over four to six weeks with combined wearable sensing and ecological momentary assessment (EMA) self-reports—could provide valuable ecological validity and enable systematic comparison between LLM-derived recommendations and observed behavioral patterns. Expanding the range of modalities beyond the current scope is also crucial. Nutrition, work environment factors (including ergonomics and screen time), socioeconomic context, and environmental exposures such

as light and noise represent particularly salient extensions, each with documented relevance for stress regulation, sleep, and well-being. Their inclusion would enhance personalization and ecological validity of recommendations.

At the same time, ethical considerations such as transparency, privacy, and the mitigation of over-reliance must be systematically addressed. Human-in-the-loop oversight can be structured so that experts shape the evaluation rubrics while end-users retain agency to accept, reject, or adapt AI suggestions. This layered approach maintains AI as the lead analytic agent but embeds essential checkpoints of human judgment. Procedurally, safeguards should include clear disclaimers that outputs are not medical advice, transparent documentation of data provenance, opt-in consent for data use, and privacy-preserving storage. Technically, accountability can be supported through audit trails and logging of evaluator decisions, allowing independent review of system behavior.

With these measures in place, embedding LLM-based recommendation systems into digital health and personal informatics platforms could provide substantial benefits in helping individuals navigate the demands of increasingly flexible and complex working environments.

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

# Reproducibility Statement

We have taken several steps to support reproducibility of this work. The study relies on a simulated seven-day multimodal dataset, which is illustrated in section 3 ("Analysis of the Potential of LLM-Based Lifestyle Adjustment Recommendations") and can be readily reconstructed by other researchers. All prompt designs and evaluation procedures are explicitly reported to enable replication. The large language model used was GPT-5, accessed through the standard web interface available to all users at the time of writing. No proprietary fine-tuning or hidden system settings were applied. While the exact outputs of generative models may vary slightly across runs, the reproducibility of the analysis lies in the transparent description of input data, prompts, and evaluation criteria.

The prompt used for generating the initial draft was as follows:

```
You are a researcher.  Write a scientific paper in English language.
Only use existing peer-reviewed scientific literature to write the
paper.  The length should be approximately 24,000 characters
including spaces.  Topic:  Potential of LLM-generated lifestyle
adaptation recommendations based on multimodal data

  0. Abstract and author information
  1. Motivation (1 page)
     Content:  Brief justification of why the topic is important,
     e.g., due to the increasing prevalence of burnouts and
     depression as well as flexibilization of work, more work from
     home (WFH), etc.  All of this leads to the fact that people need
     to organize themselves better and actively manage their health,
     well-being, and productivity.  Employers can no longer provide
     this, especially against the backdrop of working from home.
```

Overall message and conclusion: Self-management is becoming increasingly important, but it is difficult due to the many influencing variables. Therefore, IT support is important.

2. **Related Work: Diary Studies and Personal Informatics (1 page)**
   Content: Personal Informatics and other similar forms such as diary studies have become established and are also researched under the title "Diary Study" or "Ecological Momentary Assessment" and "Ecological Momentary Intervention" in research, especially in organizational and health psychology. So far, however, everything is based on self-assessments, "objective" sensor data are used too little.
   Overall message and conclusion: Personal Informatics and diary studies, especially those with supplementary sensor data, have the potential to improve self-management by revealing favorable/unfavorable patterns that could not be identified through introspection and "a bit of self-reflection" alone.

3. **Analysis of the potential of LLM-based lifestyle adaptation recommendations (4 pages)**
   Content:

   3.1 Dataset: Generation of exemplary multimodal time series data (self-assessment and sensor data)

   3.2 Generation of recommendations including mention of the LLM prompt

   3.3 Evaluation: Another LLM, which did not generate the recommendations, is instructed to evaluate the quality of the recommendations given (i) the information about a persona, (ii) their data, and (iii) the derived recommendations. Criteria could be safety, relevance, and ease of implementation of the recommendations.

   Overall message and conclusion: It is possible to derive lifestyle adaptation recommendations from an LLM based on time series data.

4. **Conclusion and Outlook (0.5 page)**

5. **References (1 page)**


