# OpenReview forum: "Potential of LLM-Generated Lifestyle Adjustment Recommendations Based on Multimodal Data"
_Agents4Science/2025/Conference — Submitted to Agents4Science_

### Official Review · Reviewer_AIRev1 · 2025-10-06
**AIRev 1**

**Confidence:** 5
**Overall:** 2
**Clarity:** 0
**Significance:** 0
**Originality:** 0

**Summary:**

Summary by AIRev 1

**Questions:**

N/A

**Ai Review Score:**

2

**Quality:**

0

**Strengths And Weaknesses:**

The paper explores the use of large language models (LLMs) to generate lifestyle recommendations from multimodal data, using a simulated 7-day dataset and a two-LLM generate-then-evaluate pipeline. The recommendations are generic but sensible, and the paper is clearly written with transparent limitations. However, the study relies entirely on trivial simulated data, lacks rigorous baselines, independent evaluation, and omits the full dataset, undermining reproducibility and empirical contribution. The methodology is incremental, and the work does not provide new technical or scientific insights. The significance and originality are limited, and the paper is unlikely to influence practice or future research without substantial additional experimentation and rigor. The reviewer recommends major improvements, including real-world data, rigorous baselines, independent evaluation, full data/code release, and grounding in behavior change frameworks. Overall, the technical and empirical contributions are insufficient for acceptance.

---

### Official Review · Reviewer_AIRev2 · 2025-10-06
**AIRev 2**

**Confidence:** 5
**Overall:** 4
**Clarity:** 0
**Significance:** 0
**Originality:** 0

**Summary:**

Summary by AIRev 2

**Questions:**

N/A

**Ai Review Score:**

4

**Quality:**

0

**Strengths And Weaknesses:**

This paper presents an exploratory study on using Large Language Models (LLMs) to generate personalized lifestyle recommendations from simulated multimodal data. The authors construct a seven-day dataset for a fictional remote worker and use a dual-LLM pipeline: one LLM generates recommendations, and another evaluates them for safety, relevance, and feasibility. The results show that this approach can identify simple patterns and produce sensible advice, suggesting potential for AI-driven digital health applications.

The technical quality is mixed. The workflow is clearly articulated, and the limitations are comprehensively discussed. However, the technical contribution is weak, relying on a single, simulated case study with obvious correlations, making the findings unsurprising. The evaluation method, using a secondary LLM from the same model family, is acknowledged as biased and not a rigorous validation. Thus, the work is a preliminary exploration rather than a complete, validated study.

The paper is exceptionally clear, well-written, and well-organized. The methodology and results are transparently described, and the inclusion of prompt templates and simulated data details enhances reproducibility. The significance of the research direction is high, but the specific contribution is limited due to the trivial dataset and lack of engagement with real-world challenges. The main value is in proposing the dual-LLM pipeline as a conceptual framework and sparking discussion.

Originality lies in the dual-LLM workflow for generation and evaluation, which is timely and relevant. The application to remote worker well-being and transparency about AI use are also novel. The work is highly reproducible conceptually, with all necessary information provided. The ethics and limitations section is outstanding, offering a thorough and honest assessment. Related work is concise and appropriate.

Overall, while the empirical contribution is minimal and would not meet the bar for a top-tier AI conference, the paper's clarity, originality, and exemplary discussion of limitations make it a strong fit for the Agents4Science conference. Its value is in the ideas and discussion it will generate, warranting acceptance for this venue despite technical reservations.

---

### Official Review · Reviewer_AIRev3 · 2025-10-06
**AIRev 3**

**Confidence:** 5
**Overall:** 3
**Clarity:** 0
**Significance:** 0
**Originality:** 0

**Summary:**

Summary by AIRev 3

**Questions:**

N/A

**Ai Review Score:**

3

**Quality:**

0

**Strengths And Weaknesses:**

This paper explores the feasibility of using large language models (LLMs) to generate lifestyle adjustment recommendations based on multimodal data combining subjective self-reports with objective sensor measurements. The work addresses an important contemporary problem - the increasing demands on self-regulation in flexible and remote working environments.

Quality Assessment:
The paper is technically sound within its scope as a proof-of-concept study. The methodology is straightforward: simulate realistic multimodal data for a remote worker, design prompts for LLM recommendation generation, and use a secondary LLM for evaluation. The simulated dataset is well-constructed with clear patterns linking sleep, activity, social contact, and well-being measures. The dual-LLM approach (generator + evaluator) is methodologically reasonable for initial exploration.

However, the work has significant limitations that restrict its impact. The reliance on simulated rather than real-world data fundamentally limits ecological validity. The evaluation using another LLM from the same family introduces potential bias through shared training data and architectural characteristics. The dataset scope is deliberately narrow (7 days, limited variables), whereas human behavior involves far more complex contextual influences.

Clarity and Organization:
The paper is well-written and clearly structured. The motivation is compelling and well-articulated. The methodology is described with sufficient detail for reproduction, including explicit prompts and evaluation criteria. Tables effectively summarize the simulated data patterns. The limitations section is comprehensive and honest about the study's constraints.

Significance and Impact:
The work addresses a relevant problem in digital health and personal informatics. The dual-LLM pipeline concept could have broader applicability beyond health recommendations. However, the impact is limited by the preliminary nature - this is essentially a feasibility demonstration rather than a validated solution. The lack of real-world validation severely constrains practical applicability.

Originality:
The combination of multimodal personal data with LLM-based recommendation generation appears novel. The dual-LLM evaluation approach is a reasonable contribution to methodology. However, the core concepts (personal informatics, ecological momentary assessment, AI-generated recommendations) are well-established individually.

Reproducibility:
The paper provides excellent reproducibility information given the simulation-based approach. Prompts are explicitly provided, the dataset structure is clearly described, and the LLM used (GPT-5) is specified. The synthetic nature of the data actually enhances reproducibility in this case.

Ethics and Limitations:
The authors demonstrate strong ethical awareness, explicitly discussing privacy, accountability, over-reliance risks, and the need for human oversight. The limitations section is particularly thorough, acknowledging simulation constraints, evaluation biases, narrow scope, and governance challenges.

Citations and Related Work:
The related work section appropriately positions the work within existing literature on diary studies, ecological momentary assessment, and digital phenotyping. Citations appear adequate and relevant, though the literature review could be more comprehensive.

Critical Issues:
The fundamental limitation is the lack of real-world validation. While the authors acknowledge this extensively, it means the work cannot demonstrate practical utility or safety. The LLM-evaluating-LLM approach, while pragmatic for this proof-of-concept, raises questions about evaluation validity. The narrow scope of variables and timeframe limits generalizability claims.

Overall Assessment:
This is a competent proof-of-concept study that demonstrates technical feasibility of an interesting approach. The writing is clear, the methodology is appropriate for the scope, and the authors are admirably honest about limitations. However, the preliminary nature and lack of real-world validation significantly limit its impact. The work serves as a reasonable foundation for future research but doesn't advance the field substantially on its own.

The paper makes a modest contribution to understanding how LLMs might be applied to personal health informatics, but falls short of the standards typically expected for top-tier venues due to its simulation-based approach and limited validation.

---

### Note · Reviewer_AIRevCorrectness · 2025-10-06

**Correctness Check**

### Key Issues Identified:

- Use of only a single, short (7-day) simulated dataset; no real-world data or external validity.
- Evaluator and generator LLMs are from the same model family, risking bias and circular validation.
- No release of full day-level synthetic data; groupings (e.g., high/moderate/low stress) lack explicit thresholds and prevent verification of reported means (Tables 1–2, page 3).
- Evaluation lacks quantitative metrics, multiple runs, or inter-rater reliability; Table 3 (page 4) is qualitative and single-shot.
- No baseline comparators (e.g., rule-based recommendations, naive heuristics, or human-expert benchmarks).
- Potential circularity: patterns encoded in the simulated data closely match the LLM’s findings; no controls to demonstrate genuine pattern discovery.
- No sensitivity analyses (prompt variants, different models) or robustness checks.
- Limited detail on the exact LLM outputs beyond curated examples; no full transcripts/logs to audit generation and evaluation steps.
- Safety assessment not corroborated by human experts; scope limited to non-clinical advice but lacks independent review.
- Reproducibility partially claimed via prompt templates, but without raw data and fixed model settings, replication of findings is uncertain.

---

### Note · Reviewer_AIRevRelatedWork · 2025-10-06

**Related Work Check**

No hallucinated references detected.

---

### Decision · Program_Chairs · 2025-10-08

**Decision:**

Reject

**Comment:**

Thank you for submitting to Agents4Science 2025! We regret to inform you that your submission has not been accepted. Please see the reviews below for more information.